# Structure and Performance Optimization of Co Magnetic Thin Films Deposited by Vacuum Evaporation Coating

**DOI:** 10.3390/ma16093395

**Published:** 2023-04-26

**Authors:** Mingheng Mao, Shaoqiu Ke, Dingguo Tang, Xiahan Sang, Danqi He

**Affiliations:** 1State Key Laboratory of New Materials and Composites Technology, Wuhan University of Technology, Wuhan 430070, China; 303516@whut.edu.cn (M.M.); shaoqiuke@whut.edu.cn (S.K.); xhsang@whut.edu.cn (X.S.); 2School of Chemistry and Materials Science, Central South University for Nationalities, Wuhan 430074, China; tdgpku@mail.scuec.edu.cn

**Keywords:** magnetic films, annealing, vacuum evaporation coating, remanence ratio

## Abstract

Co magnetic films are widely used in high-frequency magnetic recording and vertical magnetic recording due to their high saturation magnetization and magnetocrystalline anisotropy. In this work, ferromagnetic Co magnetic films were prepared on copper substrate by vacuum evaporation combined with heat treatment (H_2_ atmosphere), to investigate the impact of film thickness and annealing temperature on microstructure and magnetic properties. The results show that with the increase in annealing temperature, the Co thin film physical phase does not change significantly, the crystallinity increases, and the grain size increases, which is consistent with the results obtained from the SEM morphology map of the sample surface, leading to an increase in coercivity. By annealing experiments (atmospheric atmosphere) on Co magnetic films with and without an Al protective layer, as shown by scanning electron microscopy microscopic characterization results, it was verified that the Al layer can protect the inner Co layer from oxidation. As the film thickness increases from 10 to 300 nm, the magnetic properties of Co films change significantly. The saturation magnetization gradually increases from 0.89 to 5.21 emu/g, and the coercivity increases from 124.3 to 363.8 Oe. The remanence ratio of the 10 nm magnetic film is 0.82, which is much higher than the film remanence ratio of 0.46 at 50 nm. This is because when the thickness of the film is between 10 and 50 nm, the magnetic moments partially deviate from the in-plane direction, and the out-of-plane component reduces the film remanence ratio. This study shows that optimizing annealing temperature and film thickness can effectively control the structure and magnetic properties of Co magnetic films, which is of great significance for the development of the magnetic recording field.

## 1. Introduction

Magnetic materials are important functional materials that are widely used for energy conversion [1,2], data sensing [3], data processing [4], and information storage [5,6,7]. Magnetic thin film materials and devices are especially important for information storage [8,9,10,11,12,13]. Ferromagnetic films have a large magnetization rate, which means that they can be magnetized to saturation under a small external magnetic field (10–100 Oe), and maintain their magnetization after the removal of the external magnetic field. Above room temperature, pure metallic elements such as Fe, Co, Ni, and Gd are ferromagnetic, among which Co has the highest Curie temperature up to 1130 °C. Ferromagnetic Co films are widely used in high-frequency magnetic recording [14] and perpendicular magnetic recording [15,16] due to their high saturation magnetization strength, high-temperature ferromagnetism, and magnetic crystal anisotropy characteristics. In order to further improve the magnetic recording density, it is important to further regulate the microstructure and magnetic property of Co magnetic thin films. Magnetic thin film preparation techniques are crucial for the microstructure and magnetic properties of thin films. At present, the main techniques for the preparation of magnetic thin films include molecular beam epitaxy [17,18], magnetron sputtering coating [19,20,21], and vacuum evaporation coating [22,23]. Among them, molecular beam epitaxy and magnetron sputtering require ultra-high vacuum, precise air pressure control, and temperature control system. Therefore, these systems are usually large and costly, with extremely low growth rates, and low sample preparation efficiency. During vacuum evaporation coating, the powder material under vacuum conditions is quickly heated up by loading a large current, and then melted, evaporated, and deposited on the substrate to form a thin film material. The preparation process is simple, with high preparation speed and high efficiency.

Currently, many researchers have investigated Co magnetic thin films. Co magnetic thin films were deposited on a Mgo substrate using magnetron sputtering; to avoid Co film oxidation, a layer of Cr was used to be the buffer layer [24]. Moreover, some researchers have used magnetron sputtering to prepare Ti and Co-based metal and oxide thin film coatings and discussed the effect of Co content variation on the magnetic properties of the films [25]. Considering the effect of temperature on magnetic films, polycrystalline Co2FeGe magnetic films were successfully prepared by the magnetron sputtering technique, and the effect of postdeposition annealing temperature on structural, static, and dynamic magnetic properties was systematically investigated [26]. Recently, it was reported that the geometric parameters of magnetic films were precisely controlled by constructing nanoporous alumina substrates, and finally, CoO/Co magnetic films were prepared by an evaporative coating technique [27].

In this paper, Co magnetic films were prepared on copper foil substrate by vacuum evaporation coating combined with annealing (H_2_ atmosphere). The influence of thickness and annealing temperature on the microstructure and magnetic properties of Co films was systematically investigated. It is found that varying the Co layer thickness and annealing temperature can modulate the structure and static magnetic properties of the magnetic films in a wide range, and finally, the possible future applications of this research are presented.

## 2. Materials and Methods

### 2.1. Film Preparation

The preparation process of Co magnetic thin film is shown in Figure 1a, including two stages: vacuum evaporation coating and tube furnace annealing. Firstly, a series of Co magnetic films (with Al layer) of different thicknesses (10, 50, 100, 200, 300 nm) and Co magnetic films (without Al layer) with Co layer thickness of 200 nm were deposited on the copper foil substrate by vacuum evaporation coating. Highly pure Co powders (≥99.9%, Aladdin Company, Beijing, China) and Al powders (≥99.9%, Aladdin Company) were used to synthesize Co magnetic film. Then, the precursor films were deposited on cooper substrates by evaporating Co and Al powders. The masses of Co and Al powders are 1.15 g (Co layer 100 nm) and 0.25 g, respectively, and we obtained a magnetic thin film with a Co layer thickness of 300 nm by preparing three equal parts of Co powders (1.15 g) and placing it on three evaporation boats. During the evaporation process, the working pressure was maintained at 5 × 10^−4^ Pa, the current applied for Co powders and Al powders was controlled in the range of 190–210 A and 130–150 A. The thickness was controlled by using a calibrated quartz crystal device during the deposition process, and the vaporization rate was maintained at 0.7 nm/s to control the film thickness. Secondly, the as-deposited precursor films were annealed in a tube furnace at different temperatures (100 °C, 200 °C, 300 °C, 400 °C) for 2 h. The base pressure was lower than 0.02 MPa. The atmosphere we used was a mixture of H_2_ and Ar with a ratio of 1:9. We removed the air from the tube furnace by passing Ar atmosphere for 1 h before heating to ensure the safety of the experiment. Co magnetic films were naturally cooled to room temperature with the furnace after the end. Finally, light-gray Co magnetic films were obtained (Figure 1b).

### 2.2. Structural Characterization and Performance Testing

The phase composition of the film was characterized by the Rigaku Smartlab SE X-ray diffractometer (XRD) produced by Rigaku in Japan (Tokyo), and the specific test conditions were Cu target Kα radiation, the wavelength λ was 1.5406 Å, the working voltage was 40 KV, the working current was 50 mA, and the acquisition angle was 30–90°. SU8020 field emission scanning electron microscopy (FESEM) produced by Hitachi of Japan ( Tokyo) was used to observe the microscopic morphology of the film surface and analyze the grain size change. The samples were cut into 3 × 3 cm sizes and fixed with conductive glue on a special sample-making table, and finally, the surface morphology was characterized by VG Multilab 2000 X-ray (Waltham, MA, USA) photoelectron spectrometer (XPS) that was used to analyze the fine Co 2p spectrum of thin films before and after annealing. The magnetic properties (M-H hysteresis loop) of the sample are measured by the Versalab multifunctional vibrating sample magnetometer system (VSM) produced by Quantum Design (San Diego, SC, USA). The sensitivity of the VSM instrument was characterized by the standard deviation of the magnetic moment of the empty sample rod, which can reach 10^−7^–10^−8^ emu. The specific test conditions are temperature of 300 K, maximum applied magnetic field of 2 T, and rise/fall rate of 200 Oe/sec. The obtained data were corrected for the magnetic signal from the substrates and the sample holder.

## 3. Results and Discussion

Figure 2 shows XRD patterns of Co magnetic films (10, 50, 100, 200, and 300 nm) of different thicknesses prepared by the vacuum evaporation coating process. All samples have diffraction peaks around 44.4°, which coincide with the Co standard card JCPDS 04-020-5482 characteristic peak, which corresponds to the (111) characteristic diffraction peak of Co. As the film thickness increases, the diffraction peak of Co gradually increases. According to the Scheller formula D = k λ/β cosθ (D is the grain size, k is the Scherrer constant, β is the half-peak height width of the sample diffraction peak, θ is the Bragg angle, and λ is the X-ray wavelength). The calculation results of average grain size are shown in Table 1. The grain size increases with the increase in Co layer thickness.

The in-plane hysteresis loop of Co magnetic films of different thicknesses is shown in Figure 3a, and it can be seen that as the thickness of the Co layer increases from 10 to 300 nm, the film content increases, and the saturation magnetization intensity of the Co film gradually increases from 0.89 to 5.21 emu/g (Figure 3b). Figure 3c shows the normalized hysteresis loop of Co magnetic films of different thicknesses. It can be seen from the figure that when the thickness is 10 nm, the residual magnetic ratio of Co film is 0.82, and when the thickness is increased to 50 nm, the residual magnetic ratio of the film is only 0.46, which is reduced by 44% compared with 10 nm Co film. As the film thickness increases, the magnetic moment of some magnetic domains deviates from the in-plane direction of the film, producing a component in the out-of-plane direction [28], resulting in a decrease in the remanence ratio of the film [29,30]. Therefore, when the thickness of the Co film continues to increase, the residual magnetic ratio of magnetic films is still low. The relationship between coercivity and different thicknesses of Co magnetic films shows that with the increase in film thickness, the coercivity gradually increases, and the maximum value is 375.1 Oe at a thickness of 200 nm (Figure 3d).

To investigate the effect of heat treatment temperature on Co magnetic films, we annealed Co films with a thickness of 200 nm in a tube furnace for 2 h at different temperatures. Samples annealed at 100 °C, 200 °C, 300 °C, and 400 °C are labelled as A1, A2, A3, and A4, respectively, and the unannealed sample is labeled as A0. Figure 4a shows the XRD pattern of Co magnetic films before and after annealing at different temperatures. It can be seen from the figure that the Co characteristic diffraction peaks of all films coincided with the Co standard card JCPDS 04-020-5482 characteristic peaks, and the characteristic diffraction peak angle is around 44.4°, and no other diffraction peaks appear. With the increase in annealing temperature, the diffraction peak of Co gradually increased, indicating that the crystallinity of the film increased with the increase in heat treatment temperature. The enlarged view of the characteristic diffraction peak of Co (Figure 4b) shows that the characteristic diffraction peak of Co shifts to a high angle with the increase in annealing temperature, which can be seen from the Bragg equation 2dsinθ = n λ (d is the crystal plane spacing, θ is the angle between the incident X-ray and the corresponding crystal plane, n is the diffraction series, and λ is the wavelength of X-rays). Therefore, the crystallographic plane spacing of Co becomes smaller due to the increase in heat treatment temperature, resulting in the shift of the diffraction peak to a high angle. We also found that as the annealing temperature increases, the half-peak height and width of the diffraction peak slightly decrease, which may be caused by the larger grain size of the Co film, according to the Scheller formula D = k λ/β cosθ [31] (D is the grain size, k is the Scherrer constant, β is the half-peak height width of the sample diffraction peak, θ is the Bragg angle, and λ is the X-ray wavelength). As the grain size is inversely proportional to the half-height and width of the diffraction peak, we speculate that as the annealing temperature increases, the thin film grain grows (Table 2). Therefore, the half-peak height and width of the diffraction peak are reduced in the XRD test results, and combined with the SEM topography (Figure 4c–e), it can be seen that some grains grow with the increase in heat treatment temperature. Figure 4f–h show the EDX element analysis of Co magnetic thin film, indicating a uniform distribution of elements.

In order to determine the reduction effect of H_2_ ambient annealing on Co magnetic films, Co 2p fine spectra of Co magnetic films were tested before and after annealing at different temperatures using X-ray photoelectron spectroscopy. Figure 5 shows the Co 2p fine spectra of magnetic films A0, A1, A2, A3, and A4. From the figure, it can be seen that after annealing in the reducing atmosphere H_2_ environment, the Co 2p_1/2_ peak and Co 2p_3/2_ peak move to the direction of low binding energy from A0 to A3, indicating that there is electron transfer from Co in the films after annealing, probably due to the oxidation of the inner Co films vaporized with trace amounts of Co, and after annealing, these oxidized Co are reduced by H_2_, and from A3 to A4, the Co 2p_1/2_ peak and Co 2p_3/2_ peak and Co 2p_3/2_ peak position remained unchanged from A3 to A4, and all the inner layer Co was reduced.

From the above X-ray photoelectron spectroscopy analysis, it was concluded that the inner Co layer of the Co magnetic thin film was partially oxidized. In order to study the oxidation resistance of Co magnetic thin film, Co thin film (without Al protective layer) was additionally prepared on copper foil substrate by evaporative coating as a control group. The Co magnetic films and Co films (without Al protective layer) were annealed in a muffle furnace at 200 ℃ with an annealing time of 30 min, and Figure 6 shows the surface microscopic morphology of the samples before and after annealing. As shown in Figure 6a,b, the surface peritectic morphology of the Co film (without Al protective layer) changed greatly after annealing, in other words, the Co on the surface of the film was oxidized. As shown in Figure 6c,d, there is no significant change in the surface microscopic morphology of the Co magnetic film before and after annealing, and the Al layer on the surface of the film can effectively protect the inner Co layer from oxidation; i.e., the prepared Co magnetic film has excellent oxidation resistance.

Figure 7a is the in-plane hysteresis loop of Co magnetic film before and after annealing at different temperatures, while Figure 7b is the relationship between the saturation magnetization intensity of Co magnetic film and the annealing temperature. It can be seen that compared with the unannealed film, the saturation magnetization intensity of Co magnetic film after annealing is significantly increased, which is due to the oxidation of trace Co in the evaporated Co magnetic film. After annealing by reducing atmosphere, the oxidized trace Co is reduced. Figure 7c shows the relationship between the coercivity of Co magnetic film and the annealing temperature. The coercivity of the film gradually increases with the increase in annealing temperature, from 336.8 Oe without annealing to 471.4 Oe (after annealing at 400 °C), which is caused by the growth of the grain size after annealing.

In order to investigate the magnetic anisotropy effect of heat treatment on Co magnetic films, we measured the normalized hysteresis loop of the film by applying magnetic fields in different directions, as shown in Figure 8. Figure 8 a,b show normalized hysteresis loops with the applied magnetic field direction perpendicular and parallel to the film surface, respectively. After the annealing of Co magnetic films at different temperatures, the normalized hysteresis loop does not change significantly, indicating that heat treatment does not induce anisotropic changes in Co magnetic films. Compared with the applied magnetic field parallel to the surface direction of the film, when the applied magnetic field is perpendicular to the surface direction of the film, the saturation and applied magnetic field strength of the Co magnetic film is smaller, indicating that the easy magnetization axis is biased to the vertical film surface direction, and the difficult magnetization axis is biased to the parallel film surface direction.

## 4. Conclusions

A series of Co magnetic films with different thicknesses (10 nm, 50 nm, 100 nm, 200 nm, and 300 nm) were prepared using vacuum evaporation coating combined with tube furnace heat treatment (H_2_ environment) process, and the effects of thickness and heat treatment temperature on the magnetic properties of Co magnetic films were discussed.

(1)When the film thickness was increased from 10 to 300 nm, the XRD results showed that the physical phase of the Co magnetic film did not change. The hysteresis line results showed that the magnetic properties of Co films changed significantly, the saturation magnetization strength and coercivity increased gradually, and the remanence of Co magnetic films at 10 nm was 0.82, which was much higher than that of films at 50 nm.(2)Co magnetic films and Cu/Co (without an Al protective layer) films were prepared by vacuum evaporation coating process, and then placed in a muffle furnace for annealing experiments (atmospheric environment). SEM images of the surface of the films before and after annealing showed that the outer Al layer in the Co magnetic films effectively protected the inner Co layer from oxidation.(3)To study the effect of heat treatment temperature on the Co magnetic films, we annealed the Co films with a thickness of 200 nm in a tube furnace at different temperatures for 2 h. The annealing temperatures were 100 °C, 200 °C, 300 °C, and 400 °C (H_2_ atmosphere), respectively. With the increase in annealing temperature, the XRD results showed that there was no significant change in the physical image of Co magnetic films, the half-peak width of Co characteristic diffraction peaks gradually decreased, and the grains gradually grew up; SEM morphology showed that the grains of Co magnetic films gradually grew up with the increase in annealing temperature, which was the same as the calculated results in XRD; the results of X-ray photoelectron spectroscopy showed that there was oxidation of Co in the inner layer of Co magnetic films, which was reduced after the annealing heat treatment. The hysteresis results show that the saturation magnetization intensity of Co magnetic films increases with the increase in annealing temperature, while the Co magnetic films have obvious magnetic anisotropy, with the magnetization axis biased toward the vertical plane of the films, and the annealing heat treatment does not induce the anisotropy change in the films.

This study shows that optimizing the annealing temperature and film thickness can effectively regulate the structure and magnetic properties of Co magnetic films. So far, this study of Co magnetic films is based on commercial copper foil with an Al layer as a protective layer, which has not been reported, and which provides a new development idea for future applications.

## Figures and Tables

**Figure 1 materials-16-03395-f001:**
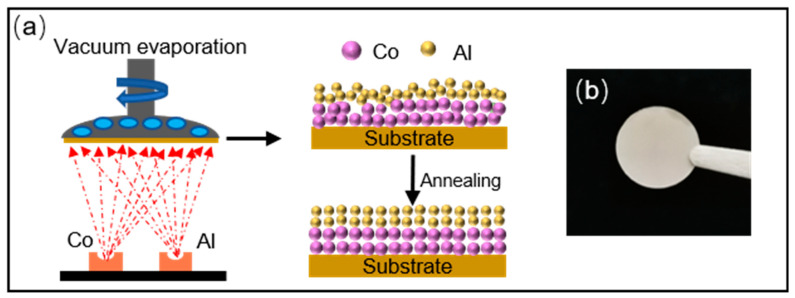
(**a**) Schematic diagram of magnetic film prepared by vacuum evaporation; (**b**) Physical drawing of magnetic film.

**Figure 2 materials-16-03395-f002:**
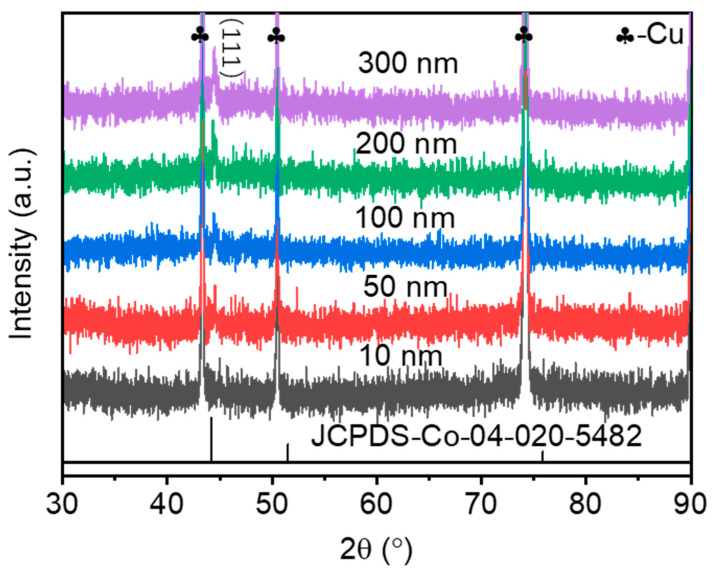
XRD patterns of Co magnetic films with different thicknesses.

**Figure 3 materials-16-03395-f003:**
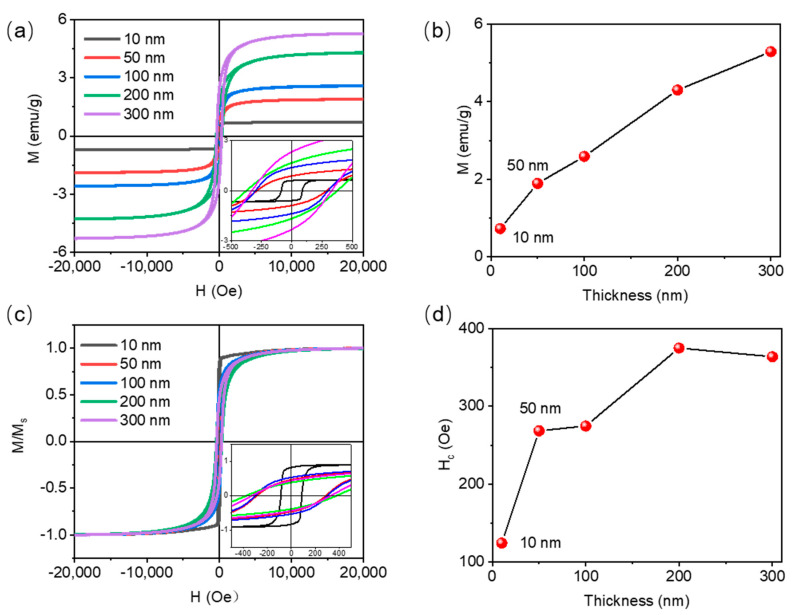
(**a**) In-plane hysteresis loop of Co magnetic films with different thicknesses; (**b**) Relation curve between saturation magnetization and thickness of Co magnetic films; (**c**) In-plane normalized hysteresis loop of Co magnetic films with different thicknesses; (**d**) Relation curve between coercivity and thickness of Co magnetic films.

**Figure 4 materials-16-03395-f004:**
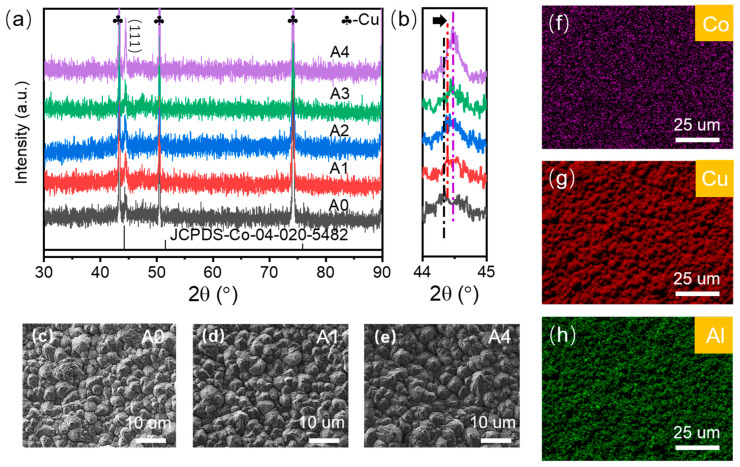
(**a**) XRD patterns of Co magnetic films before and after annealing at different temperatures; (**b**) XRD patterns in the range of 44–45°; (**c**) SEM image of Co magnetic film before annealing; (**d**) SEM image of Co magnetic film annealed at 100 °C; (**e**) SEM image of Co magnetic film annealed at 400 °C; (**f**–**h**) EDX image of Co, Cu, Al elements.

**Figure 5 materials-16-03395-f005:**
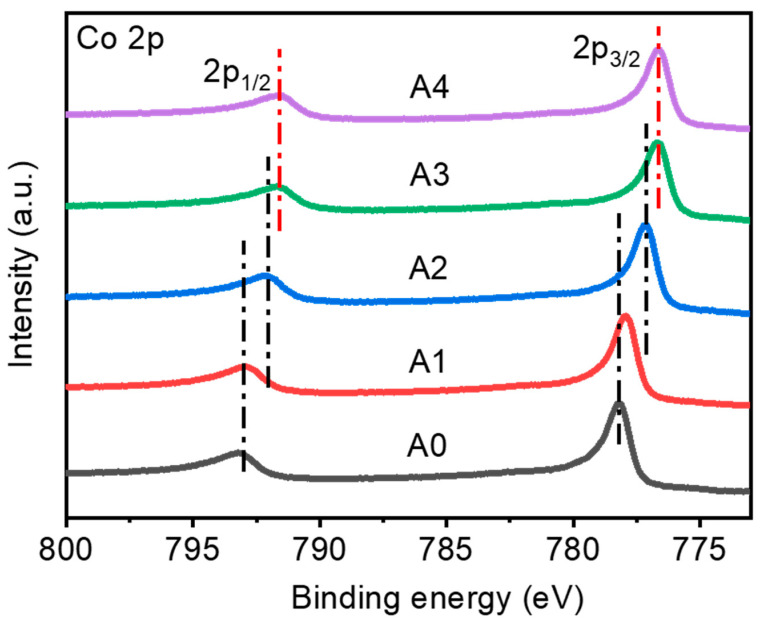
Fine Co2p spectra of X-ray photoelectron spectroscopy of Co magnetic thin films annealed at different temperatures.

**Figure 6 materials-16-03395-f006:**
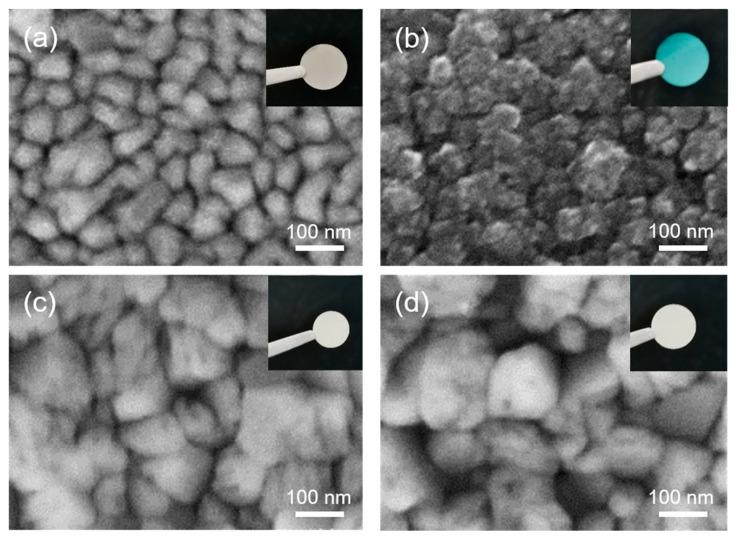
(**a**) SEM diagram of Co film before annealing, (**b**) SEM diagram of Co film after annealing, (**c**) SEM diagram of Co/Al magnetic film before annealing, (**d**) SEM diagram of Co/Al magnetic film after annealing.

**Figure 7 materials-16-03395-f007:**
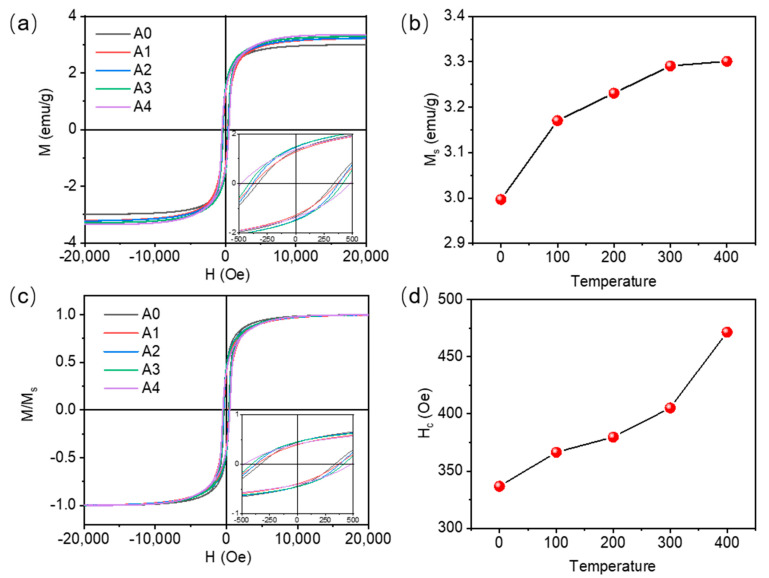
(**a**) In-plane hysteresis loops of Co magnetic thin films before and after annealing at different temperatures; (**b**) relationship between saturated magnetization strength of Co magnetic thin films and annealing temperature; (**c**) normalized in-plane hysteresis loops of Co magnetic thin films before and after annealing at different temperatures; (**d**) relationship between coercivity of Co magnetic thin films and annealing temperature.

**Figure 8 materials-16-03395-f008:**
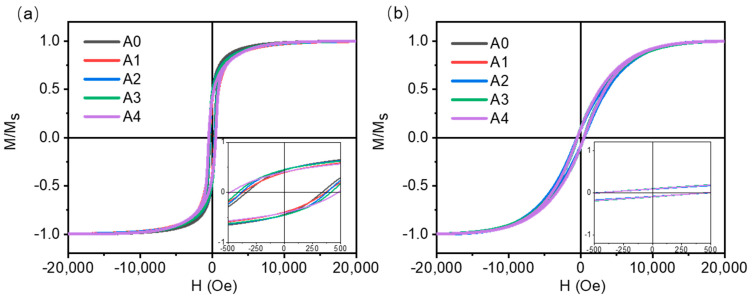
Normalized hysteresis loops of Co magnetic films applied in different directions with external magnetic field: (**a**) the direction of the magnetic field is perpendicular to the surface of the film, (**b**) the direction of the magnetic field is parallel to the surface direction of the film.

**Table 1 materials-16-03395-t001:** The average grain size (D_1_) varies with thickness.

Thickness (nm)	10	20	100	200	300
D_1_ (nm)	8 ± 2	12 ± 2	23 ± 3	31 ± 3	39 ± 3

**Table 2 materials-16-03395-t002:** The variation in average grain size (D_2_) with annealing temperature when the thickness of Co layer is 200 nm.

Temperature (°C)	0	100	200	300	400
D_2_ (nm)	31 ± 3	41 ± 3	65 ± 3	97 ± 3	137 ± 3

## Data Availability

Research data are not shared.

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
