# Peer review of "Structure and Performance Optimization of Co Magnetic Thin Films Deposited by Vacuum Evaporation Coating"

_materials, 2023, doi:10.3390/ma16093395_

Round 1

Reviewer 1 Report

The manuscript discusses the thermal evaporation of Co films for their magnetic properties. The films of various thicknesses are deposited and annealed in the presence of H2. Many experimental details are missing. What was the base pressure? How muh H2 concentraion is used for annealing? What precautions are taken to use H2 at such high temperatures? On its own, the work is ok, but when compared with the literature following things should be clarified

1.    Thermal evaporation deposition of Co film is not new. There are so many publications dealing with it. However, none of the work is mentioned in the introduction section of the manuscript.

2.    The obtained results are not compared with the results of the literature for its properties.

3.    Why is there so much shift in the Co 2p peaks in the XPS with the increase in thickness

4.    What is shown in the inset of Figure 6?

5.    How to understand the erratic behaviour of Figure 3d.

6.    How to confirm that there is no oxygen in the film?

7.    Does the Aluminum coating shields the magnetic properties, or how does it affects and when it is annealed at 400 degrees centigrade, what happens to it?

Line 206: I think you meant with instead of after.

Line 28 it shoud be for 

Ok to a certain extent, still the use of preposition is not good.

Reviewer 2 Report

The current work under title Structure Regulation and Performance Optimization of Co Magnetic Thin Films, illustrated the possibility to obtain Co-thin films with different layer thickness. The magnetic and microstructure analysis are presented. I feel the outcome result of the work and its interesting is in average. I recommend to accept the work after the authors make a major revision in all the manuscript. Following the comment in details;

1-      The introduction is very poor and need to enhanced with more recent advanced in Co thin films prepared with different physical form such as Co thin films in (dots, strips, wires, microwires and antidots) here are some suggested references to include in the introduction parts:

•    Salaheldeen, M.; Nafady, A.; Abu-Dief, A.M.; Díaz Crespo, R.; Fernández-García, M.P.; Andrés, J.P.; López Antón, R.; Blanco, J.A.; Álvarez-Alonso, P. Enhancement of Exchange Bias and Perpendicular Magnetic Anisotropy in CoO/Co Multilayer Thin Films by Tuning the Alumina Template Nanohole Size. Nanomaterials 2022, 12, 2544. https://doi.org/10.3390/nano12152544
•    Wojcieszak, D.; Mazur, M.; Pokora, P.; Wrona, A.; Bilewska, K.; Kijaszek, W.; Kotwica, T.; Posadowski, W.; Domaradzki, J. Properties of Metallic and Oxide Thin Films Based on Ti and Co Prepared by Magnetron Sputtering from Sintered Targets with Different Co-Content. Materials 2021, 14, 3797. https://doi.org/10.3390/ma14143797.

2-      Materials and Methods

The authors have to include more details about the deposition process of the Co films. In addition, the authors have to describe how to measure the layer thickness.

3-      Results and discussion;

·         The XRD analysis must be enhanced with the average grain size estimation for as prepared and annealed Co thin films with different layer thickness in a table. The changing in the average grain size plays an important role in the magnetic properties of samples.

·         The authors have to include the uncertainties for the Estimations of layer thickness, saturation magnetization (Figure 3b), coercivity (Figure 3d) and the same for magnetic parameters plotted in Figure 7.

·          SEM images in Figure 4 needs to presented in a clear way and it will be great if you add the EDX mapping to illustrate the elements distribution.

·         The references need to be enhanced with more recently articles related to the topic.

Reviewer 3 Report

In this study, author have prepared a Co magnetic film was prepared by vacuum evaporation coating combined with annealing for investigating the impact of its thickness  and annealing temperature on microstructure and magnetic properties. The film was then characterized in terms of XRD, SEM, XPS and VSM.

However, authors have to address below comments:

·         The FESEM result should be included in Abstract.

·         The sample preparation method should be mentioned in method section for methods like FESEM.

·         Revise the fig. captions (like Fig.3 captions) in term of “space” among words.

·         Curves in Fig.3 B is no obvious.

·         The vertical space between (a) and (c) as well as (b) and (d) in images of Fig.3 is not enough.

·         The characteristic peaks before and after annealing of film (Fig.4) should be mentioned in text.

·         Some word are capital like “Annealed” in Fig.4 caption, which must be corrected.

·         “XPS” or “X-ray photoelectron spectroscopy” should be mentioned in Fig.5 caption.

·         RSD of results must be mentioned for quantitative analysis.

·         There is no enough references for “results and discussion”. It should be enriched.

·         Furthermore, literature survey and comparison with similar studies are required.

·         Author should address the novelty and advantages of the study (specially compared to “https://doi.org/10.3390/nano11051229” and “https://doi.org/10.1016/j.micron.2005.10.013” and mention the limitation and future perspective of the study.

·         The text requires editing of English language.

Best

·         Revise the fig. captions (like Fig.3 captions) in term of “space” among words.

·     Some word are capital like “Annealed” in Fig.4 caption, which must be corrected.

The text requires editing of English language.

Best

Round 2

Reviewer 1 Report

What happens to the Al thin fil when annealed at 400 degree C is not answered?

need a bit improvement

Reviewer 2 Report

The author's fixed all my comments and I recommend to accept the manuscript in a present version. 

Author Response

Dear Reviewer,

We would like to thank  the reviewer for the assistance in revising and publishing the paper.

Kind regards,

Reviewer 3 Report

.

.

Author Response

Dear reviewer,

We would like to thank  the reviewer for the assistance in revising and publishing the paper. The whole manuscript has been polished, therefore, the readers may understand our work more clearly.

Kind regards,